# An Experimental Study on the Vibration Transmission Characteristics of Wrist Exposure to Hand Transmitted Vibration

Mingzhong Wu *, Sheng Jia and Zhihong Lin

College of Mechanical Engineering and Automation, Huaqiao University, Xiamen 361021, China; jiasheng_na@foxmail.com (S.J.); lin123hongzhi@163.com (Z.L.)
* Correspondence: jdwmz62@hqu.edu.cn

**Abstract:** This research intends to further improve the understanding of vibration damage mechanisms in the wrist area and to establish a more effective biodynamic model of the hand-arm system. Scholars have conducted some research work around the influencing factors of vibration response and commonly used vibration transmissibility to characterize the local vibration transmission characteristics of the hand-arm system. In this paper, a hand-transmitted vibration test platform was built according to ISO 10819, and a random combination of four ergonomic factors, namely wrist posture, arm posture, grip force, and thrust force, was used to test the vibration response of six subjects' wrists; the total vibration transmissibility of the wrist was calculated according to the transmissibility formula. The effect of the four factors on the total vibration transmissibility of the wrist part was comprehensively analyzed, in which the wrist posture was proposed for the first time. The results show that (1) vibration transmissibility of the wrist is not only related to the arm posture, thrust force, and grip force but also related to the wrist posture; (2) the total vibration transmissibility and resonance frequency on the wrist has small correlation with large grip force and thrust force, and the vibration transmissibility of grip force 30 N and 60 N are basically equal in the low-frequency band (from 5–10 Hz to 5–20 Hz); (3) the wrist postures have a significant effect on the total vibration transmissibility at the wrist.

**Keywords:** wrist posture; hand-transmitted vibration; vibration transmissibility; arm posture; grip force; thrust force

## 1. Introduction

Hand-arm vibration disease has become a common occupational disease in industrial and agricultural countries [1,2]. Hand-arm vibration disease is usually caused by long-term hand-transmitted vibration work, which mainly involves peripheral circulation disorders of the hand, nerve dysfunction of the arm, and bone and joint-muscle damage to the arm; the typical manifestation is vibration white finger [3]. The white finger phenomenon is the main basis for the diagnosis of hand-arm vibration disease, and the damage to the hand-transmitted vibration to the hand-arm system, it is not just limited to the finger area. Studies have shown that wrist injuries or symptoms such as carpal tunnel syndrome (CTS) are associated with hand-transmitted vibrations [4–7], grinding with wrists in an improperly bent position, which is the main cause of CTS in the wrist area [8]. Although studies have confirmed epidemiologically and clinically that wrist injuries are associated with hand-transmitted vibrations, the pathogenic mechanism is not yet clear. The study of the vibration transmission properties of the wrist is an important basis for understanding the mechanism of its vibration disorder. Currently, experimental studies on the vibration transmission characteristics of hand-arm systems are mainly conducted based on ISO 10819; however, ISO 10819 specifies only one posture of the wrist (W1, as shown in Figure 1a) [9]. ISO 5349-1 states that the posture of the wrist, elbow, and shoulder will also affect the

transmission of vibration response [10]. In actual working scenarios, the wrist posture is diverse due to the different power tools, workplace, and work nature, etc. This study hopes to better understand the vibration transmission characteristics of the wrist part and the hand-transmitted vibration damage mechanism of the wrist part. There are many researchers who study the effect of arm posture on the response characteristics of the hand-arm system [11,12], all of which mainly consider the effect of elbow flexion angle (also called arm posture, as shown in Figure 2) on the response characteristics, while the effect of wrist posture is not considered. Xu et al. studied the vibration transmission characteristics of the wrist under a specific posture in a real working environment [13]. Xu et al. also investigated the vibration transmissibility characteristics on a specific posture of holding the horizontally mounted handle with both hands; none of the effects of other wrist postures were considered, and the effect of push force on vibration transmissibility is analyzed only in [14]. Besa et al. investigated the effect of three wrist postures on the mechanical impedance characteristics of the whole hand-arm system in an experimental environment; the study considered a wrist posture different from the one in this paper, and the effect of push force is not considered [15]. Adewusi et al. investigated the vibration transmission characteristics of the wrist part in the *z*-axis and *y*-axis directions on the standard posture specified by ISO 10819. Other postures of the wrist were not considered, and vibration transmissibility in single axial direction is analyzed only in [16].

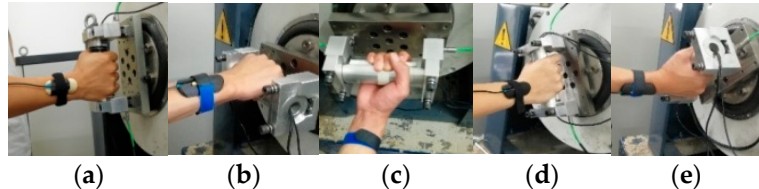

**Figure 1.** Wrist posture. (**a**) W1, (**b**) W2, (**c**) W3, (**d**) W4, (**e**) W5.

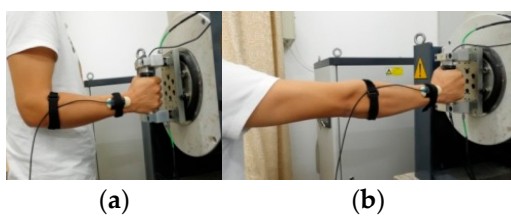

**Figure 2.** Hand-arm posture. (**a**) SP1, (**b**) SP2.

In summary, this research hopes to further improve the understanding of vibration damage mechanisms in the wrist region and to establish a more effective biodynamic model of the hand-arm system. In this study, the effects of arm posture, grip force, thrust force, and wrist posture on the total vibration transmissibility of the wrist region are comprehensively analyzed, among which the effect of wrist posture is proposed for the first time. The different wrist postures are obtained by holding the vibrating handle at the different mounting angles (as shown in Figure 3: (a) 90°, (b) 0°, (c) 30°, and (d) 45° indicate that the handle is mounted vertically, horizontally, vertically inclined to the left at 30° and vertically inclined to the right at 45° respectively). In this paper, the vibration response of the wrist part in three orthogonal directions is obtained by experiment, then the total vibration transmissibility of the wrist part is also obtained. The total vibration transmissibility of the wrist on W2, W3, W4, and W5 postures (as shown in Figure 1b–e), is compared with the total vibration transmissibility in W1 standard posture, and the total vibration transmissibility at the wrist site under the influence of grip force, thrust force, and arm posture are compared with the reported vibration response characteristics.

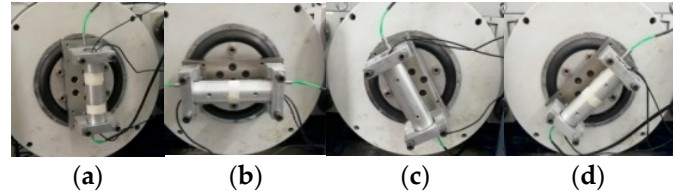

| (a) | (b) | (c) | (d) |

**Figure 3.** Handle on different angle. (**a**) 90°, (**b**) 0°, (**c**) 30°, (**d**) 45°.

## 2. Methods

### 2.1. Experimental Set-Up

The test system for measuring the hand-transmitted vibration response of the hand-arm system is built according to ISO 10819:2013 [9] (as shown in Figures 4 and 5). The test system consists of a single-axis electromagnetic shaker, a vibration controller, and an instrumental handle (as shown in Figure 6), of which the instrumented handle has a diameter of 40 mm and a length of 120 mm (the length of the grip part), the middle section of the handle is cut in half along the axis to form the handle base and the handle cover, and the inside of the handle is hollowed out to install a three-axis acceleration sensor (Dytran 3023A2, sensitivity: 10 mv/g) and two force sensors (ZNHBM161217) which measure the grip force. The handle bracket is connected to the transducer (Kistler 9317 C) for measuring thrust force and is fixed to the electromagnetic shaker by a clamp. The axis of the instrumental handle mounted on the handle bracket is perpendicular to the ground, and the vibration controller drives the electromagnetic shaker to vibrate along *z*-axis direction according to the given excitation signal. The *z*-axis is parallel to the ground and is perpendicular to the centerline of the handle (*y*-axis direction), as shown in Figure 4.

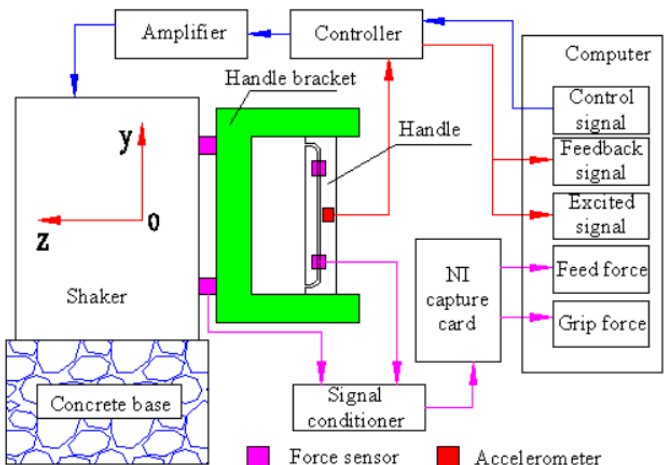

**Figure 4.** Schematic diagram of test system.

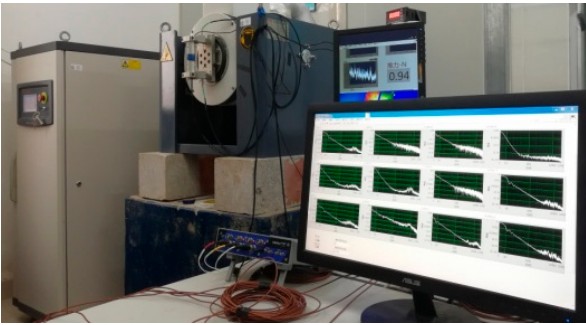

**Figure 5.** The test system.

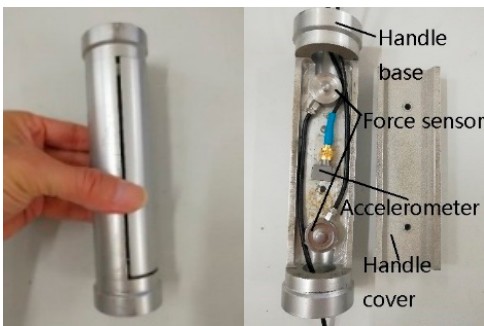

**Figure 6.** Instrumented handle.

The three-axis acceleration sensor in the handle base is installed in the middle of the handle base and the three orthogonal measurement directions of the sensor are aligned with the coordinate system of the hand (as in Figure 7). The sensor is used to measure the z direction acceleration that is the vibration signal of the single-axis electromagnetic shaker. The excitation signals used in the experiment are consistent with those specified in ISO 10819:2013 [9]. Broadband random vibration in the frequency range of 5 to 1600 Hz is shown in Figure 8, with the *x*-axis being the frequency (f), the *y*-axis being the power spectral density (PSD), the reference signal being the excitation signal set by the controller, and the control signal is the actual feedback excitation signal. The sensor for measuring acceleration at the wrist is fixed using an adapter; one of the directions of the sensor must be parallel to the excitation direction z of the shaker. The shape and dimensions of this adapter are determined by reference to the ISO 10819:2013 [9] and the dimensions of the sensor. The adapter obtained by 3D printing is as shown in Figure 9. The use of this adapter not only ensures that the three orthogonal measurement directions of the sensor coincide with the direction of the vibration transmitted to the hand but also increases the contact area between the sensor and the arm, reducing the influence of the adapter and the sensor on the measurement results [17].

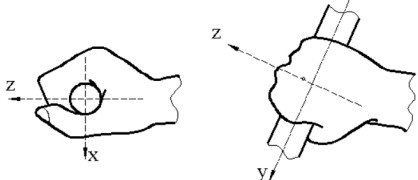

**Figure 7.** Coordinate system of the hand.

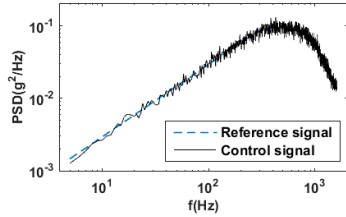

**Figure 8.** Excited signal.

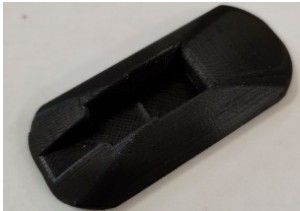

**Figure 9.** Accelerometer adapter.

## 2.2. Subjects and Measurement Steps

The trial is testified on six subjects, all six of whom are male. The anthropometric data and age of the subjects are shown in Table 1, and the mean in the table is the mean of the six subjects and SD is the standard deviation. Before the test, the subject learns the steps of the test and is instructed to hold the handle in the specified posture and keep the grip and thrust force within the specified values. The subject is trained several times before the test to ensure that reliable test data can be obtained during the test. The test is conducted with the subject standing upright on a horizontal floor using three different grip forces (g0 N, g30 N, g60 N) and three different thrust forces (p20 N, p50 N, p80 N), and the magnitude of the grip and thrust forces are displayed in real-time by a display during the test. The wrist posture and arm posture used in the measurement are shown in Figures 1 and 2, and the elbow should not touch the body during the measurement. The sensor on the wrist is fixed in the adapter and then tied to the wrist area by elastic straps, which are tightened to the extent that the subject feels comfortable, and the sensor is always fixed to the backside of the wrist in different wrist postures, as in Figure 1.

**Table 1.** Anthropometry data of the subjects in the vibration experiments.

| Subject | Height (cm) | Weight (kg) | Hand Length (mm) | Hand Width (mm) | Hand Volume (mL) | Age (n) |
|---|---|---|---|---|---|---|
| A | 168.0 | 60.6 | 180 | 85 | 305 | 25 |
| B | 171.5 | 78.4 | 195 | 94 | 380 | 26 |
| C | 181.0 | 66.1 | 192 | 83 | 320 | 24 |
| D | 173.5 | 73.3 | 192 | 88 | 345 | 23 |
| E | 181.5 | 76.0 | 200 | 93 | 390 | 25 |
| F | 170.5 | 51.7 | 185 | 88 | 285 | 24 |
| Mean | 174.3 | 67.68 | 190.7 | 88.5 | 337.5 | 24.5 |
| SD | 5.15 | 9.34 | 6.52 | 3.95 | 38.2 | 0.96 |

Note: hand length = tip of middle finger to crease at the wrist, hand width = the width measured at metacarpal of the hand, hand volume = volume of water discharged when the hand is immersed in water.

Each subject was required to repeat the measurement three times under the same test conditions, and a total of 1620 sets of data ($6 \times 5 \times 2 \times 3 \times 3 \times 3$) were obtained in this experiment; the test conditions are randomly combined and the same subject is guaranteed sufficient rest time between trials. The duration of each experiment is at least 30 s. In the experiment, NI 9234 is used for signal acquisition with a sampling rate of 4000 Hz, and the initial processing of data is completed by Labview programming.

## 2.3. Vibration Transmissibility

The definition of transmissibility indicates the response ratio before and after the vibration of the power tool passes through the hand-arm system, and the transmissibility can understand the decay of the vibration through the hand-arm system [18]. The vibration transmissibility $T$ in a certain direction is the ratio of the root-mean-square acceleration $A_{hand-arm}$ (denoted as $A_h$) in one direction of the hand or arm to the root-mean-square acceleration $A_{reference}$ (denoted as $A_r$) at the same direction of the reference point on the handle, i.e., $T = A_h/A_r$ [19]. The vibration transmissibility in a certain frequency range is calculated by $T(f_i) = A_h(f_i)/A_r(f_i)$, where $f_i$ is the center frequency of the $i$th one-third octave band, $A_h(f_i)$ and $A_r(f_i)$ are the acceleration at the center frequency $f_i$ of the 1/3 octave band of a part of the hand-arm and the handle reference point, respectively, and $T(f_i)$ is the vibration transmissibility corresponding to the center frequency $f_i$ of the $i$th one-third octave band.

In fact, most of the tools transmit vibration in three axes, and when the single-axis shaker used in the test is coupled with the human hand-arm, vibration is also transmitted along the non-axial direction. In addition, the thrust force in the non-axial direction often leads to considerable vibration in the other two directions. To solve the problems

of misalignment of the sensor installation and the non-axial loading of the thrust force that causes large errors which will exceed 20%, the tri-axial acceleration sensor is used to measure the acceleration of the handle and a part of the hand-arm system in three orthogonal directions. The vibration transmissibility of each axial direction is calculated according to Equation (1), and the total vibration transmissibility (*T1*) of the three axial directions is calculated according to Equation (2) [19].

$$T_i = A_{hi}/A_{ri} \dots i = x, y, z \tag{1}$$

$$T1 = \frac{\sqrt{A_{hx}^2 + A_{hy}^2 + A_{hz}^2}}{\sqrt{A_{rx}^2 + A_{ry}^2 + A_{rz}^2}} \tag{2}$$

where $A_{hi}$ is the root-mean-square acceleration in the *x*, *y* and *z* directions of a part of the hand-arm, and $A_{ri}$ is the root-mean-square acceleration in the *x*, *y* and *z* directions of the handle reference point, i.e., the input acceleration. The transmissibility is not specifically stated below; refer to the total vibration transmissibility *T1*.

## 3. Results and Discussion

In this section, Figures 10–12 represent the effects of arm posture, grip force, and thrust force on *T1* at the wrist site in W1 posture, respectively. Each curve in Figures 10–12 shows the mean values of the transmissibility of the six subjects under different test conditions.

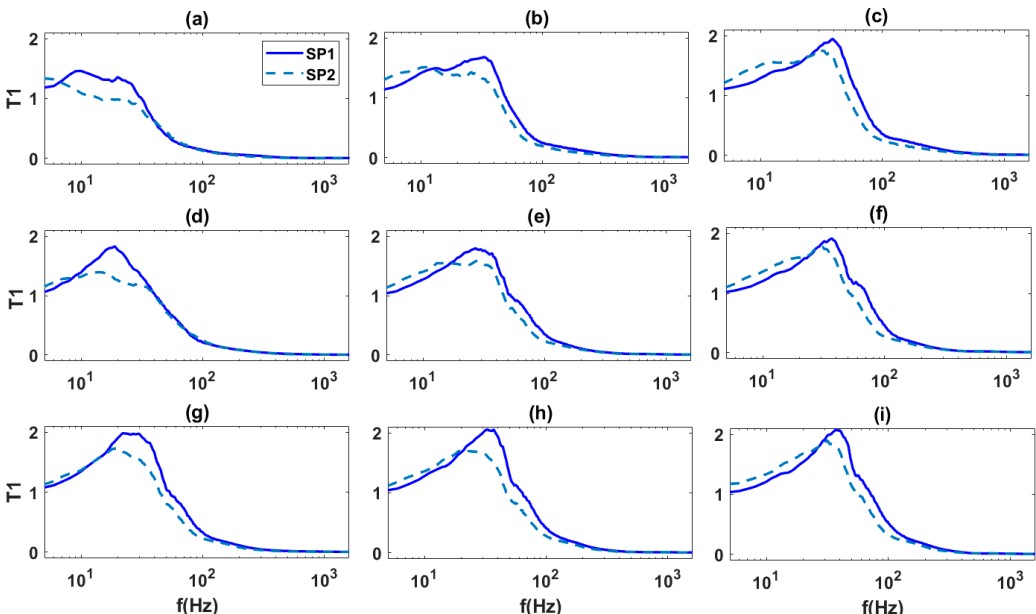

**Figure 10.** Effect of arm posture on the averaged total vibration transmissibility. (**a**) g0 N, p20 N, (**b**) g0 N, p50 N, (**c**) g0 N, p80 N, (**d**) g30 N, p20 N, (**e**) g30 N, p50 N, (**f**) g30 N, p80 N, (**g**) g60 N, p20 N, (**h**) g60 N, p50 N, (**i**) g60 N, p80 N.

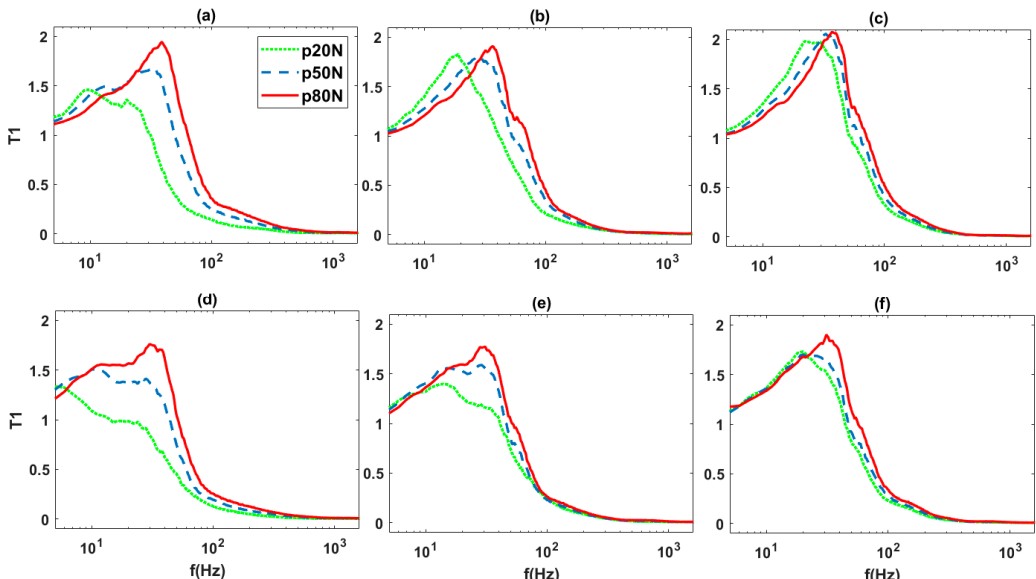

**Figure 11.** Effect of the thrust forces on averaged total vibration transmissibility. (**a**) SP1, g0 N, (**b**) SP1, g30 N, (**c**) SP1, g60 N, (**d**) SP2, g0 N, (**e**) SP2, g30 N, (**f**) SP2, g60 N.

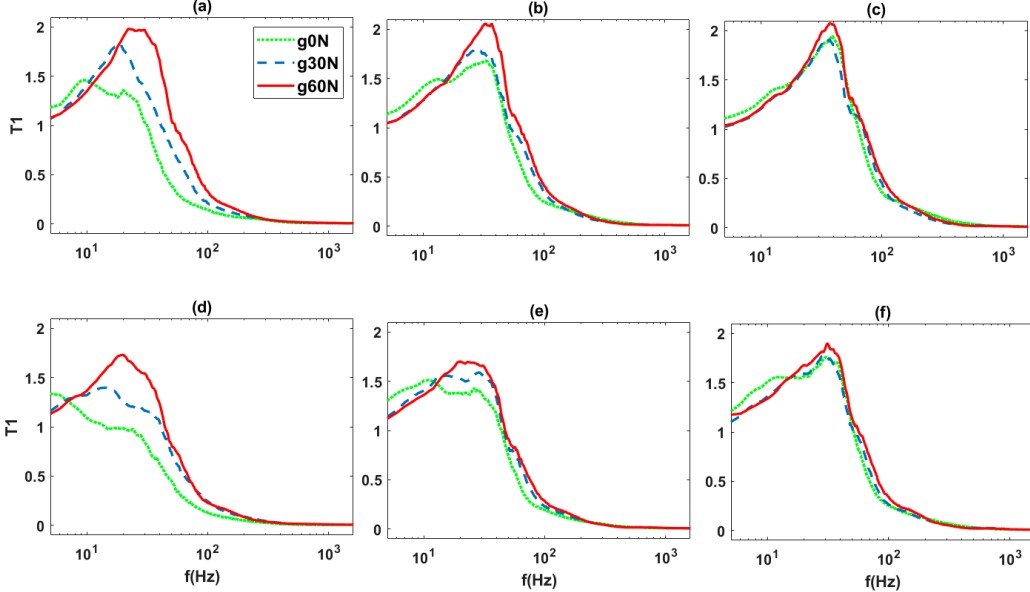

**Figure 12.** Effect of grip force on the total averaged vibration transmissibility. (**a**) SP1, p20 N, (**b**) SP1, p50 N, (**c**) SP1, p80 N, (**d**) SP2, p20 N, (**e**) SP2, p50 N, (**f**) SP2, p80 N.

As seen in Figures 10–12, the trends of the transmissibility curves are the same. Before the resonance peak, the transmissibility for all of them increases with the increase of frequency, and the hand plays an amplifying role in the transmitted vibration. Furthermore, after the resonance peak, the transmissibility decreases rapidly with the increase of frequency, and the hand plays an attenuating role in the transmitted vibration. Therefore, the effect of the hand on the vibration transmission characteristics is non-linear.

### 3.1. Effect of Arm Posture

Figure 10 shows the effect of arm posture on the transmissibility under nine different combinations of grip force and thrust force. From Figure 10, we can see that the effect of arm posture on the transmissibility is different, the resonance peak of the wrist part under SP1 posture is larger than that of SP2 posture, and the effect of arm posture on the resonance

peak values becomes smaller as the grip force and thrust force increase. Arm posture also affects the resonance frequency of the transmissibility. In the case of the same grip force and thrust force, the resonance frequency of SP1 posture is always greater than that of SP2 posture; the figure also shows that, left of the intersection of the two transmissibility curves, the vibration transmissibility of the wrist part under the SP1 posture is less than or close to that of the SP2 posture. At the right of the intersection point, the vibration transmissibility of the wrist part under SP1 posture is greater than that of SP2 posture, and as the grip force or thrust force increases, this intersection point shifts to the right, that is, the frequency corresponding to the intersection point increases. In the case of small grip or thrust force, double resonance peaks appear in both curves, and with the increase of thrust force and grip force, the double resonance peaks in both curves disappear in succession. The study in the literature [14] shows that only the elbow angle 120° arm posture shows the double peak, but the SP1 posture does not express the double wave peak, which is different from the study in this paper. The difference is caused by the different grip and thrust force in a different experiment, and another possibility is that individual data averaging may cause double peaks to be inconspicuous or disappear.

The trend of the vibration transmissibility on the wrist region is similar to the results of the literature [15,16,20], but the literature [16] only analyzes the characteristics of the vibration transmission of the bent arm and extended arm postures in the *z*-axis direction at a thrust force of 50 N and grip forces of 10 N, 30 N, and 50 N. The literature [15] studies the impedance characteristics in the *z*-direction on the SP1 and SP2 postures (80 N grip force and 0 N thrust force). The literature [20] studies the vibration transmission characteristics of the fingers at various points with the open hand placed on the shaker table. The experimental conditions of those studies differ significantly from those of this paper, and the effect of arm posture on the vibration transmission characteristics is not analyzed in the literature [20].

### 3.2. Effect of Thrust Force

Figure 11 shows the effect of thrust force on the vibration transmissibility of the wrist. As can be seen from Figure 11, the trend of the vibration transmissibility of the wrist part is similar under different grip forces and arm posture, and the resonance frequency increases correspondingly with the increase of the thrust force, while the resonance peak value increases with the increase of the thrust force. The trend of the increase of the resonance peak value is related to the size of the grip force: the smaller the grip force the faster the increase of the resonance peak value, and the larger the grip force the slower the increase of the resonance peak value. At the grip force of 60 N, the resonance peak increases slightly with the increase of the thrust force. The figure also shows that left of the intersection point, the vibration transmissibility of the wrist part under the SP1 posture decreases with increasing thrust force, while the vibration transmissibility of the wrist part under SP2 posture decreases slightly with increasing thrust force.

The trend of the effect of thrust force on the transmissibility at the wrist site is the same as that of the literature [14], and the resonance peak value at the wrist site in the literature [14] increases slightly with increasing thrust force; this difference may be due to the relatively small thrust force (20 N), which was used in this experiment, and the relatively large thrust force (100 N) used in the literature [14]. The two transmissibility curves in Figure 11c for a thrust force of 50 N and 80 N are also particularly close, which is consistent with the results of the literature [21] (transmissibility is almost constant when grip and thrust forces reach a certain value).

### 3.3. Effect of Grip Force

Figure 12 shows the effect of different grip forces on the vibration transmissibility of the wrist part. As can be seen from Figure 12, the effect of grip force on the vibration transmissibility of the wrist part under six different test conditions is similar to that of the thrust force. The resonance peaks value and peaks frequency of the vibration transmissi-

bility of the wrist part increase with the increase of the grip force, but this changing trend is related to the size of the thrust force; the frequencies of the resonance peaks and peaks value increase significantly with the increase of the grip force (as shown in Figure 12a,d, while at the thrust force increases from 20 N to 50 N and 80 N. The trend of the resonance peak increases with the increase of the grip force and the frequency of the peak tends to be constant (as shown in Figure 12b–f).

The results of this study are similar to those of study [15] and study [16]; Ref. [15] shows that below 15 Hz, impedance does not change with grip force and the difference appears in the resonance frequency, and [16] shows that within the frequency 5–15 Hz, grip force has almost no effect on vibration transmissibility of the wrist in the *z*-axis directions, and above 15 Hz, vibration amplitude and resonance frequency increase with grip force increase. As seen from the six plots in Figure 12, the vibration transmissibility under grip force 30 N and 60 N is basically equal in the low frequency band for the same arm posture and thrust force, which is consistent with the results of the literature [15,16], but there is some difference in the frequency range where they equal, and the frequency band where the vibration transmissibility of grip force 30 N and 60 N are equal in this study varies from 5–10 Hz to 5–20 Hz. This should be related to the arm posture and the size of the thrust force. In addition, the difference between the vibration transmissibility under grip force 0 N (i.e., with only thrust force) in the low frequency band and that of grip force 30 N and 60 N is significantly larger than that between grip force 30 N and 60 N.

*3.4. Effect of Wrist Posture*

Each subplot in Figures 13 and 14 represents the vibration transmissibility of the wrist part under 5 different wrist postures, and the 9 subplots correspond to 9 different combined of grip forces and thrust forces; Figure 13 shows the SP1 arm posture and Figure 14 shows the SP2 arm posture. From the overall view of Figure 13, with the change of grip force and thrust force, there is no obvious regularity in the change of the relative magnitude of the vibration transmissibility of the five wrist postures under different grip force and thrust force, although the trend of the change of the vibration transmissibility under different wrist postures is similar. As seen in Figure 14, except for Figure 14d, the W5 resonance peaks are larger than those of the other four wrist postures. In order to more precisely analyze the differences in vibration transmissibility of the five wrist postures, the results of comparing the peak value of vibration transmissibility of the other four wrist postures with that of the W1 posture are summarized in Table 2.

As seen in Table 2, the peak vibration transmissibility of W1 posture is larger than that of other postures when the grip force and thrust force are small, except for the grip force of 0 N and thrust force of 20 N; the upper right corner of Table 2 is "+", which indicates that the peak vibration transmissibility of W1 is smaller than that of other wrist postures when the grip force and thrust force are larger in SP1 posture. Most of the lower right corners of Table 2 are "-" and "~", indicating that in the SP2 posture, the peak vibration transmissibility of W1 is greater than or equal to that of the other wrist postures but less than that of the W4 posture. In conclusion, if the vibration damage of various wrist parts is evaluated in W1 posture, it may be overestimated or underestimated; therefore, the wrist posture should be considered when evaluating the vibration damage of wrist parts.

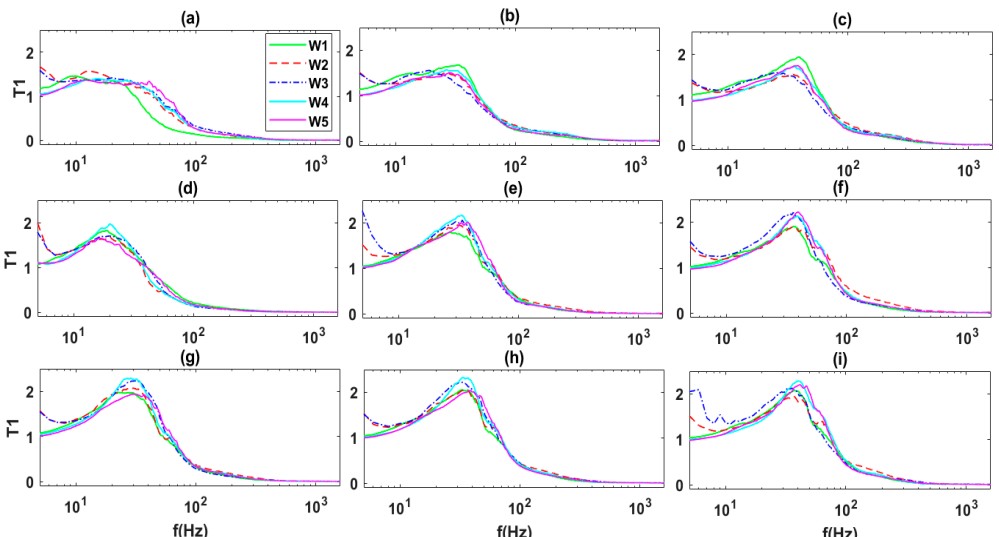

**Figure 13.** The vibration transmissibility at the wrist part on different wrist posture (SP1). (**a**) g0 N, p20 N, (**b**) g0 N, p50 N, (**c**) g0 N, p80 N, (**d**) g30 N, p20 N, (**e**) g30 N, p50 N, (**f**) g30 N, p80 N, (**g**) g60 N, p20 N, (**h**) g60 N, p50 N, (**i**) g60 N, p80 N.

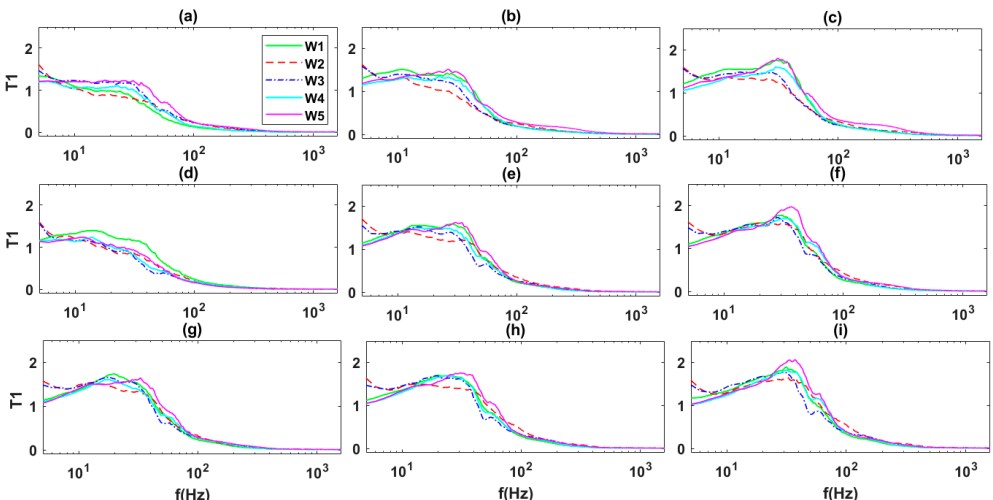

**Figure 14.** The vibration transmissibility at the wrist part on different wrist posture (SP2). (**a**) g0 N, p20 N, (**b**) g0 N, p50 N, (**c**) g0 N, p80 N, (**d**) g30 N, p20 N, (**e**) g30 N, p50 N, (**f**) g30 N, p80 N, (**g**) g60 N, p20 N, (**h**) g60 N, p50 N, (**i**) g60 N, p80 N.

**Table 2.** Comparison of the transmissibility of the wrist on the other four wrist postures to the wrist1 posture under different arm posture, grip force, and push force.

| Arm Posture | Wrist Posture | Grip Force and Thrust Force | | | | | | | | |
|---|---|---|---|---|---|---|---|---|---|---|
| | | g0 N p20 N | g0 N p50 N | g0 N p80 N | g30 N p20 N | g30 N p50 N | g30 N p80 N | g60 N p20 N | g60 N p50 N | g60 N p80 N |
| SP1 | W2 | + | - | - | ~ | + | + | + | + | + |
| | W3 | + | - | - | - | + | + | + | + | + |
| | W4 | + | - | - | - | + | + | + | + | + |
| | W5 | + | - | - | - | + | + | + | + | + |
| SP2 | W2 | - | - | - | - | - | - | ~ | - | - |
| | W3 | + | - | - | - | - | - | - | - | - |
| | W4 | + | - | - | - | ~ | + | + | + | + |
| | W5 | + | - | - | - | - | - | - | ~ | - |

Note:'+' means the transmissibility wave-peak of other wrist postures is higher than that of W1 posture, '-' means the transmissibility wave-peak of other wrist postures is lower than that of wrist1 posture, '~' means the transmissibility is approximately equal to that of wrist1 posture.

*3.5. The Subject Variability*

Figure 15 shows the total vibration transmissibility of the six subjects and their mean values, and Figure 16 shows the mean and standard deviation of the total vibration transmissibility of the six subjects. These results were obtained under grip force 30 N and thrust force 50 N with the W1 and SP1 posture. From Figures 15 and 16, it can be seen that the trends of the total vibration transmissibility of the six subjects were similar, but there were significant differences between these individuals around the peak transmissibility. The largest deviation of the transmissibility is 0.316, which occurred at the frequency of 47 Hz, and the mean value of the vibration transmissibility at this frequency is 1.23. This deviation and the mean value of the transmissibility are very comparable to the literature [16], but there is a difference of 14.2 Hz in the corresponding frequency. In addition, it can be seen from Figure 15 that each transmissibility curve has a distinct peak, with resonance frequencies between 25 and 37 Hz for the six subjects and a mean frequency of 27 Hz, all of these frequencies being within 16–50 Hz [13]. The large differences in individual vibration transmissibility may be related to their kinetic properties and anthropometric data, as well as changes in grip force, thrust force, and arm posture during the experiment [16]. Therefore, the experiment conditions should be strictly controlled and the influence on the test data should be reduced during the experiment.

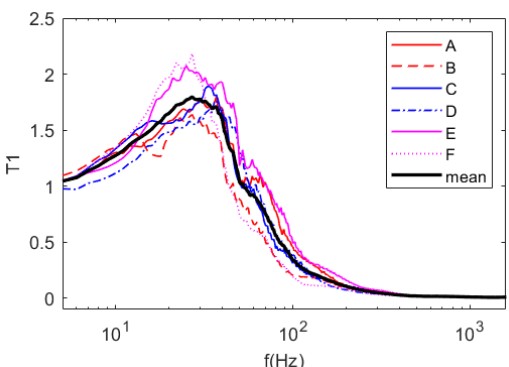

**Figure 15.** Vibration transmissibility and mean value of 6 subjects.

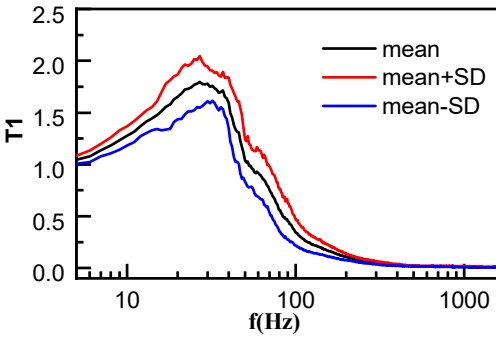

**Figure 16.** Mean and standard deviation of 6 subjects.

## 4. Conclusions

In this paper, the effects of five wrist postures are presented for the first time on the total vibration transmissibility of the wrist part, and the effects of the main influencing factors of the vibration transmissibility of the wrist part are also analyzed more comprehensively on the vibration transmission characteristics of the wrist part. These findings further enrich the hand-transmitted vibration transmission characteristics of the wrist.

Studies have shown that the vibration transmissibility of the wrist area is not only related to the arm posture, thrust, and grip force but also to the wrist posture. Before the intersection of the transmissibility curves, the vibration transmissibility of SP1 posture

was less than or close to that of SP2 posture; the vibration transmissibility of SP1 posture is greater than that of SP2 after the intersection. The total vibration transmissibility of the wrist part increases with the increase of thrust force, the resonance frequency, and the resonance peak amplitude increase. Before the intersection point, the vibration transmissibility of SP1 posture decreases with the increase of thrust force, while the vibration transmissibility of SP2 posture is almost constant with the increase of thrust. The effect of grip force on the vibration transmissibility is mainly located near the resonance peak, and in the low frequency band, the change of grip force has almost no effect on the vibration transmissibility of the wrist part. The double peaks of the transmissibility curves may not be obvious or disappear due to individual data averaging. A common characteristic of the vibration transmissibility of the wrist area is that the vibration transmission characteristics of the wrist area are almost constant, while the grip and thrust force reach a certain value.

The total vibration transmissibility is used to analyze the vibration transmission characteristics of the wrist area in this paper, which not only avoids large test errors due to the deviation of the sensor measurement direction but also is consistent with the requirements of ISO 5349 for assessing hand-transmitted vibration hazards based on the total vibration value. Therefore, the data from this study can be further used for hazard assessment of hand-transmitted vibration.

The wrist site transmissibility on W2, W3, W4, and W5 postures differ significantly from those on W1 near the resonance peak, and the assessment of wrist site injuries by the model built on the response characteristics of the wrist site measured on W1 posture will result in overestimation or underestimation. Therefore, the effect of wrist posture should also be considered in the assessment of hand-transmitted vibration injuries in the workplace.

Although there are many studies on the effects of arm posture, grip force, and thrust force on the vibration transmission characteristics of the hand-arm system, there are certain differences in the results of these studies, and the factors influencing the vibration transmission characteristics of the hand-arm system are not yet completely clear. The research in this paper further refines the vibration transmission characteristics of the wrist area. In addition, the influence of anthropometric factors on the vibration transmissibility is one of the elements that needs to be studied in the future. Standardization of the test platform is necessary to reduce the variation of test data.

**Author Contributions:** Conceptualization, M.W.; methodology, M.W.; software, M.W.; validation, M.W.; formal analysis, M.W.; investigation, M.W.; resources, M.W.; data curation, S.J.; writing—original draft preparation, M.W.; writing—review and editing, M.W.; visualization, Z.L.; supervision, M.W.; project administration, M.W.; funding acquisition, M.W. All authors have read and agreed to the published version of the manuscript.

**Funding:** This research has been supported by the Quanzhou City Science& Technology Program of China (grant No. 2018Z016).

**Institutional Review Board Statement:** The study involved human subjects, but not to the extent of requiring consent or approval from the relevant authorities.

**Informed Consent Statement:** Informed consent was obtained from all subjects involved in the study.

**Data Availability Statement:** All data included in this study are available upon request from the corresponding author.

**Conflicts of Interest:** The authors declare that they have no conflict of interest.

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
