# Peer review of "An Experimental Study on the Vibration Transmission Characteristics of Wrist Exposure to Hand Transmitted Vibration"

_applsci, doi:10.3390/app12042232_

Round 1

Reviewer 1 Report

The following points should be considered for the revision of the manuscript

  1. The abstract is not explaining the background of the study. The abstract should reflect the background, research methodology, and conclusion in a concise way. Rewrite the abstract accordingly.
  2. There is a lot of unsupported text in the paper. Address the issue accordingly
  3. Minor English revision is required 
  4. Since the presented work is very good, it is very difficult to find the significance of the proposed work. Add some extra text in the conclusion and methodology section appropriately.

Author Response

Dear editors and reviewers:

First of all, thank you for your comments on "An experimental study on the vibration transmission characteristics of the wrist exposure to hand transmitted vibration", Manuscript ID: applsci-1590558.

I would like to thank you for reviewing the paper. Based on your comments and suggestions, I would like to make the following revisions on point-by-point.

Any revisions to the manuscript is marked up using the “Track Changes” function in the MS word. That is, the red font.

Comments and Suggestions for Authors from Reviewer1

Comment 1:The abstract is not explaining the background of the study. The abstract should reflect the background, research methodology, and conclusion in a concise way. Rewrite the abstract accordingly.

Answer:I have rewritten the abstract by background, research methods, and conclusions, as detailed in the revised manuscript.

Comment 2:There is a lot of unsupported text in the paper. Address the issue accordingly.

Answer:I have made changes in the text, so please criticize and correct me if there are still omissions or incorrect points.

Comment 3:Minor English revision is required

Answer:The corresponding changes have been made in the original text, please see the revised version for details

Comment 4:Since the presented work is very good, it is very difficult to find the significance of the proposed work. Add some extra text in the conclusion and methodology section appropriately.

Answer:The significance of the work in this paper has been added in the conclusion.

The above are my revision instructions based on the review comments of reviewer 1, please correct me if there is any revision error.

Reviewer 2 Report

The abstract is rather scarce. It is repeated several times what is being measured without a clear insight into the reasons for the measurement. No proper background was given. Same can be said for the introduction part as well. Even though the number of references is sufficient, there is no broad context of why is this research important. The technical details of the experiment set up are explained with sufficient detail, but there is a major flow in the selection of the subjects and later with the interpretation of the results. It is not clear why the authors decided to test six males with obviously different anthropometry details and average their results. There is no point in measuring and averaging the influence of wrist postures, arm postures, grip forces of different persons and interpreting the results of vibrational transmissibility on that basis. 

The importance of the study and the future research directions should be indicated in the conclusion.

Author Response

Dear editors and reviewers:

First of all, thank you for your comments on "An experimental study on the vibration transmission characteristics of the wrist exposure to hand transmitted vibration", Manuscript ID: applsci-1590558.

I would like to thank you for reviewing the paper. Based on your comments and suggestions, I would like to make the following revisions on point-by-point.

Any revisions to the manuscript is marked up using the “Track Changes” function in the MS word. That is, the red font.

Comments and Suggestions for Authors from Reviewer2

Comment 1:The abstract is rather scarce. It is repeated several times what is being measured without a clear insight into the reasons for the measurement. No proper background was given.

Answer:I have rewritten the abstract by background, research methods, and conclusions, as detailed in the revised manuscript.

Comment 2:Same can be said for the introduction part as well. Even though the number of references is sufficient, there is no broad context of why is this research important.

Answer:The problems in the introduction section have been revised, please see the revised version for details。

Comment 3:The technical details of the experiment set up are explained with sufficient detail, but there is a major flow in the selection of the subjects and later with the interpretation of the results. It is not clear why the authors decided to test six males with obviously different anthropometry details and average their results. There is no point in measuring and averaging the influence of wrist postures, arm postures, grip forces of different persons and interpreting the results of vibrational transmissibility on that basis. 

Answer: The ISO10819 requirements for test number and vibration transmissibility data processing are that the number of subjects should be at least 5 adults and the vibration transmissibility should be averaged. So I consider 6 subjects and average the measurements in the paper. The literature [16] is based on 6 subjects and vibration transmissibility averaging, the literature [17] is based on 5 subjects and the literature [20] is based on 10 subjects and vibration transmissibility averaging. The subjects of the above literature were also selected randomly. The literature [16] [17] [20] involves the following statistical table of subjects, and there are other literatures as well, which are not listed here. Some anthropometric factors may also affect the vibration transmissibility, and although this issue has been mentioned in the literature, the effect of anthropometric factors on vibration transmissibility has been little studied, so this is one of the elements to be studied in the future.

The following table of subject statistics is extracted from the literature [16] .

The following table of subject statistics is extracted from the literature [17] .

The following table of subject statistics is extracted from the literature [20] .

Comment 4:The importance of the study and the future research directions should be indicated in the conclusion.

Answer: The importance of the study and the future research directions has been added in the conclusion.

The above are my revision instructions based on the review comments of reviewer 2, please correct me if there is any revision error.

Round 2

Reviewer 2 Report

The authors did correct some of the issues in Abstract and in the Introduction part, but the interpretation of the results is still lacking.  In references [17] and [20] on which they are referring, transmissibility functions of all the ten subjects measured together with their corresponding mean responses were given in the figures. Also ISO 10819 specifies only one posture of the wrist, as you have mentioned in the article. If you are expanding your research you cannot invake the standard you are trying to upgrade. At least a small discussion should be written regarding this, and additional Figures showing the result for all test subjects should be given in the supplement.

Author Response

Dear editors and reviewers:

First of all, thank you once again for the second round review of the article "An experimental study on the vibration transmission characteristics of the wrist exposure to hand transmitted vibration"(Manuscript ID: applsci-1590558).

Based on your comments and suggestions, I would like to make the following revisions.

Any revisions to the manuscript is marked up using the “Track Changes” function in the MS word. That is, the red font.

Comments and Suggestions for Authors from Reviewer2(Round2)

Comment 1:The authors did correct some of the issues in Abstract and in the Introduction part, but the interpretation of the results is still lacking.  In references [17] and [20] on which they are referring, transmissibility functions of all the ten subjects measured together with their corresponding mean responses were given in the figures. Also ISO 10819 specifies only one posture of the wrist, as you have mentioned in the article. If you are expanding your research you cannot invake the standard you are trying to upgrade. At least a small discussion should be written regarding this, and additional Figures showing the result for all test subjects should be given in the supplement.

 Answer: Based on your review comments, I am revising again on the content that was not revised well the first time. I have added section 3.5 to the text, including Figures 15 and 16. Please see the revised version for details. 

 The above are my revision instructions based on the review comments of reviewer 2(round2), please correct me if there is any revision error.